# Temperature extremes of 2022 reduced carbon uptake by forests in Europe

Auke M. van der Woude [1,2,14], Wouter Peters [1,2,14] ✉, Emilie Joetzjer [3], Sébastien Lafont [4], Gerbrand Koren [5], Philippe Ciais [6], Michel Ramonet[6], Yidi Xu[6], Ana Bastos [7], Santiago Botía[7], Stephen Sitch [8], Remco de Kok[2,9], Tobias Kneuer[10], Dagmar Kubistin [10], Adrien Jacotot [11], Benjamin Loubet[12], Pedro-Henrique Herig-Coimbra [12], Denis Loustau[13] & Ingrid T. Luijkx [2]

The year 2022 saw record breaking temperatures in Europe during both summer and fall. Similar to the recent 2018 drought, close to 30% (3.0 million km²) of the European continent was under severe summer drought. In 2022, the drought was located in central and southeastern Europe, contrasting the Northern-centered 2018 drought. We show, using multiple sets of observations, a reduction of net biospheric carbon uptake in summer (56-62 TgC) over the drought area. Specific sites in France even showed a widespread summertime carbon release by forests, additional to wildfires. Partial compensation (32%) for the decreased carbon uptake due to drought was offered by a warm autumn with prolonged biospheric carbon uptake. The severity of this second drought event in 5 years suggests drought-induced reduced carbon uptake to no longer be exceptional, and important to factor into Europe's developing plans for net-zero greenhouse gas emissions that rely on carbon uptake by forests.

The year 2022 marked another year of temperature extremes in Europe. In summer, record temperatures over 40 °C in mid-latitude countries such as France, the UK, and the Netherlands[1], and water temperatures over 30 °C in the Mediterranean Sea[2] occurred. Subsequent autumn temperatures were also elevated, with mean temperature for the months of October, November and December exceeding the long-term mean by several degrees, especially in southern Europe[3]. In western Europe this extreme summer heat is often associated with so-called blocking events when stationary Rossby wave trains across the northern hemisphere keep high pressure areas in place over the European continent, diverging moisture inflow from the Atlantic ocean north- and southwards relative to its normal westerly path[4–6]. This "wave-7" blocking pattern occurs more frequently during positive phases of the North Atlantic Oscillation (NAO[7]), and is suggested to occur more frequently with increasing climate warming[4,8]. The intense droughts of 2003, 2015, 2018, and 2022 each played out under such conditions[9–13], with several studies confirming an important role for human-made climate warming[14–17]. Although painted as exceptional climate conditions in the media, these heat and rainfall patterns have a much-reduced return time of 10–15 years under current global warming[18–20], and will be part of the "new normal" of the decades to come.

[1]University of Groningen, Centre for Isotope Research, Groningen 8481 NG, The Netherlands. [2]Wageningen University, Meteorology & Air Quality Dept, Wageningen 6700 AA, The Netherlands. [3]Université de Lorraine, AgroParisTech, INRAE, UMR Silva, 54000 Nancy, France. [4]Functional Ecology and Environmental Physics, Ephyse, INRA, Villenave d'Ornon, France. [5]Copernicus Institute of Sustainable Development, Utrecht University, Utrecht, The Netherlands. [6]UMR CEA-CNRS-UVSQ, Laboratoire des Sciences du Climat et de l'Environnement, Gif sur Yvette, France. [7]Max Planck Institute for Biogeochemistry, Jena, Germany. [8]Faculty of Environment, Science and Economy, University of Exeter, Exeter, UK. [9]ICOS ERIC, Carbon Portal, Geocentrum II, Sölvegatan 12, SE-22362 Lund, Sweden. [10]Deutscher Wetterdienst, Hohenpeissenberg Meteorological Observatory, Hohenpeissenberg, Germany. [11]Sol, Agro et hydro-systèmes, Spatialisation (SAS), UMR 1069, INRAE, Institut Agro, Rennes, France. [12]Université Paris Saclay, AgroParisTech, INRAE, UMR 1402 ECOSYS, 91120 Palaiseau, France. [13]ISPA, Bordeaux Sciences Agro, INRAE, F-33140 Villenave d'Ornon, France. [14]These authors contributed equally: Auke M. van der Woude, Wouter Peters. ✉e-mail: Wouter.Peters@wur.nl

The 2022 large-scale drought and temperature anomalies we outline here thus fit a reported shift of summer climate extremes[18]. Accumulating drought experience and better national heat plans in Spain, Portugal France, Germany, Belgium, and the Netherlands limited the worst impacts —such as the 70,000 excess deaths in 2003[21]— but water shortages, shipping disruptions, wildfires, crop yield loss, and forest degradation were nevertheless widespread once again[2]. From the perspective of forestry, fire management, agriculture, biodiversity, and carbon sequestration in Europe it is of great importance to understand the impact on carbon exchange by vegetation and soils in Europe. Especially carbon dioxide removal is gaining recent attention, as the vast majority of countries included a large potential for carbon sequestration by the forestry sector in their Nationally Determined Contributions to the Paris Agreement.

This study investigates the impact of the 2022 summer drought on carbon exchange between European forests and the atmosphere, from a diverse set of ground- and space- based monitoring platforms. By placing the 2022 event in context of previous strong summer droughts, we try to answer the question whether the carbon cycle impact of the 2022 extreme drought event was an exceptional, or exemplary, situation for upcoming drought impacts on forests.

## Results

### Anatomy of the summer drought

In 2022 an area of anomalously high pressure was centred over France (see Fig. 1) much like during the record heat wave of 2003[22] and large-scale conditions resemble the diagnosed state from that event. Like in 2003, high sea-surface temperatures in the Mediterranean Sea and low winter/spring precipitation in southern Europe[23] contributed to low soil moisture levels in early summer (See Fig. 1 and Supplementary Fig. S1), likely triggering land-surface feedbacks known to exacerbate summer heat and drought[24–27]. This contrasts the more atypical 2018 drought event over northern Europe, which occurred while the NAO index was anomalously high (+1.65 from May 2018 to September 2018) pushing the stationary high pressure centre northwards towards Scandinavia[4,28]. Figure 1 shows the geopotential height anomalies in 2022 and Fig. 2 shows the areas under severe drought (3-month standardised precipitation and evaporation index (SPEI) < −1.2) in July 2018 (blue areas, 2.7 million km²) and in July 2022 (yellow and red areas, 3.0 million km²), with the blue/yellow hatched area marking the overlapping area where both droughts hit (0.8 million km²) (see Methods). A secondary centre of drought in 2022, away from the high-pressure anomaly centred over France (Fig. 1) can be distinguished over the Eastern part of Europe (Croatia, Bulgaria, Romania, and Slovenia, red contours). The temperate land-climate (Köppen class D) of this area differs from the sea-climate (Köppen class C) between the Atlantic and Mediterranean sea, and it also has a distinct land-use with extensive beech forests.

In contrast to the 2018 drought, which was preceded by a wet winter and a warm and sunny spring[28], the 2022 drought developed from already low soil moisture (SM) levels since winter (also see Fischer et al. [9]). And rather than a warm spring, the 2022 summer drought was followed by anomalously warm conditions and persisting low soil moisture in autumn. Although a clear soil moisture anomaly is seen over Europe from February onward (Fig. 1, also see Fig. S1 and Supplementary Section B), this anomaly is not outside the 2σ range for the vast majority of the drought-affected area. However, another relevant driver of heat wave impacts in summer is the combination of atmospheric excess heat and low humidity, as seen through the vapour pressure deficit (VPD). Using the ERA5 reanalysis[29] to spatially integrate over the three affected areas shown in Fig. 2, we show that JJA-2022 had the highest VPD of any of the last 20 years in 45/51/15% of the area under the central/south/east contour respectively, with JJA-2018 close behind (also see Supplementary Fig. S3). We next report the

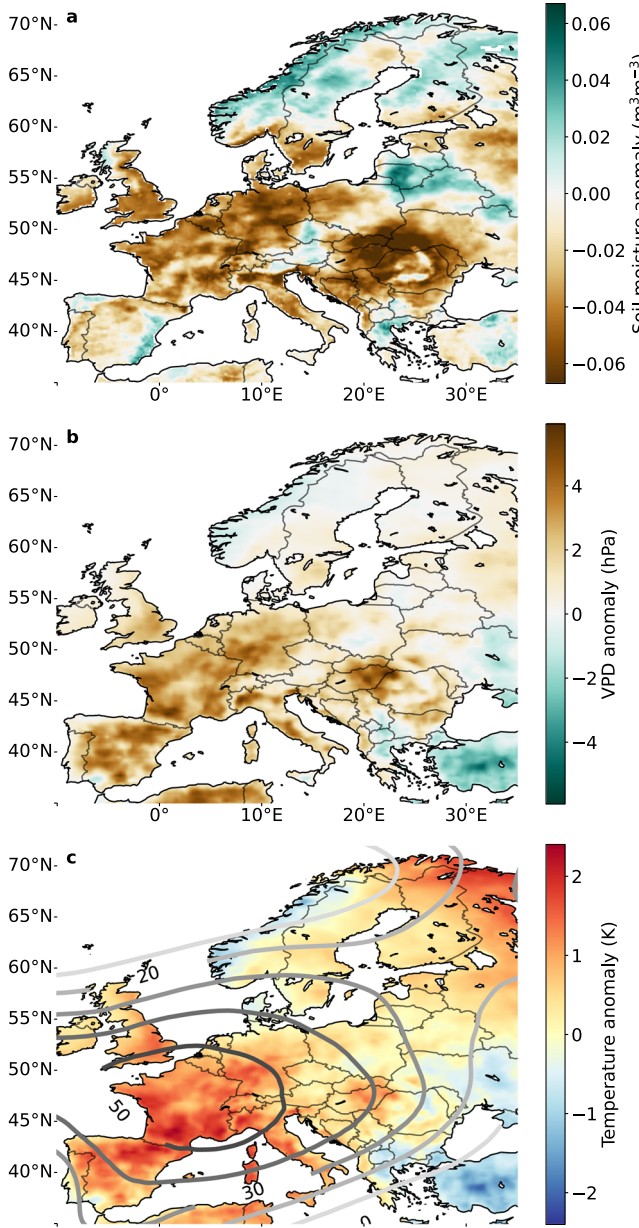

**Fig. 1 | Overview of the European drought in 2022. a** Soil moisture and **b** vapour pressure deficit (VPD) anomalies, representing soil and atmospheric drought respectively. **c** Temperature anomalies, indicating elevated temperatures over large parts of Europe. Geopotential height (GPH) anomalies (500hPa, in metres), relative to 1980–2022, for MJJA 2022 are indicated with contours. Soil moisture is taken from ERA5-Land[50], temperature, VPD and GPH are taken from ERA5[29].

widespread impacts of these effects on the carbon balance of the atmosphere and forests across Europe.

### Net carbon exchange impacts

The network of the Integrated Carbon Observing System (ICOS[30]) recorded positive anomalies in atmospheric $CO_2$ mole fractions across southern- and western Europe in near real-time, summarised in Fig. 3 (see Methods). Higher than average (2019–2021) mole fractions (see Supplementary material C) across central Europe could indicate either higher Net Ecosystem Exchange (NEE, positive for fluxes to the atmosphere, also see Methods), or a change in atmospheric circulation with advection of more northerly and $CO_2$-enriched air masses. We find, based on both observation- and model-based analyses, that the reduction in carbon uptake by the terrestrial biosphere was the

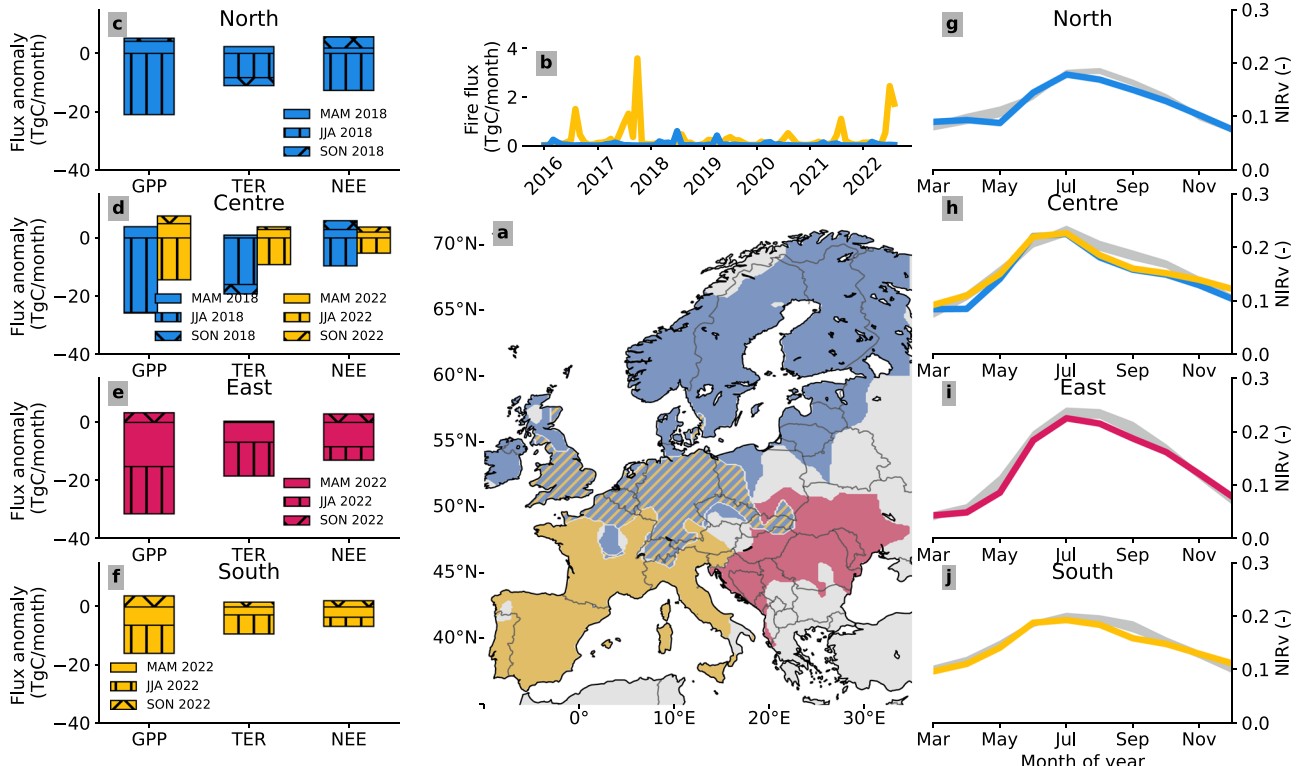

**Fig. 2 | Overview of the carbon impact of the European droughts of 2022 and 2018. a** The different regions struck by the droughts (See Methods) of 2018 ("North" in blue), 2022 ("South" in yellow, "East" in red), or both years ("Centre" in blue/yellow hatched). Note that the East region is far away from the centre of the 2022 geopotential height anomaly 1. **b** Fire fluxes, taken from GFAS[61] over the South (yellow) and North (blue) regions. **c–f** Net Ecosystem Exchange (NEE) anomalies (negative means less uptake) per region relative to 2016–2021 (excluding 2018) for MAM (clear), JJA (vertical hatching), and SON (crossed hatching), as calculated by the biosphere model SiB4 (see Methods). **g–j** Monthly mean MODIS NIRv signal (see Methods) per region for drought years, compared with the climatology between 2016 and 2021 (excluding 2018) (in grey). Colours in (**c–j**) refer to the regions of the same colour in panel a, where (**d, h**) correspond to the central region, with yellow and blue and representing 2022 and 2018, respectively.

dominant impact in July and August (±75% of the $CO_2$ signal, see Supplementary Table E), with sites in southern France showing > 2.5 ppm excess $CO_2$ in JJA, relative to 2019–2021 (Fig. 3, also see Methods and Supplementary Section C).

Biomass burning (see also Fig. 2b) contributed substantially to higher $CO_2$ mole fractions only at site Biscarosse, near the largest fires in the region of Les Landes, France, as evidenced by simultaneous increases in carbon monoxide (CO) mole fractions with 1-day averaged values exceeding 1000 ppb in early July (Supplementary Fig. S4). Below, we show a more detailed analysis of the impacts of fires. Overall, the broader spatial pattern offered by the ICOS network confirms the more southerly centre of drought impact compared to 2018, with a larger integrated atmospheric $CO_2$ summer anomaly.

A quantification of the impact on net ecosystem exchange during JJA gives a reduction of carbon uptake by vegetation of 56–62 TgC, relative to 2019–2021, over the affected areas shown in Fig. 2. This is similar to the 50–66 TgC we estimate for JJA 2018, which in turn agrees closely with our earlier estimate (49.8 TgC over the blue (including the blue/yellow hatched) area in Fig. 2) for that event[31]. The quoted range for vegetation uptake includes one mechanistic model calculation (SiB4) from our CTE-HR near real-time flux product for Europe[32], as well as first results from atmospheric inverse modeling with a limited set of observation sites (see Supplement E). The impact of fires is quantified separately below. We note that such an atmospheric inverse modeling estimate is a time-consuming task, requiring several inputs that are not directly available, and therefore inverse fluxes are typically not available until a year after such an event, while CTE-HR results are available in near real-time (one week). The close correspondence of the inverse results and the biosphere model calculations, reconfirms the capacity of the

underlying SiB4 biosphere model to convincingly capture the summer drought impact on European net ecosystem exchange, as we also reported in Smith et al. (2020, referred to as SM2020 from here on)[31].

Regionally, we find the central (blue-yellow hatched) area in Fig. 2a that was hit twice by summer droughts to have responded less strongly in JJA 2022 (an anomaly of 7.8 TgC in 2022 and 19.7 TgC in 2018) (see Fig. 2d), but also experienced at a slightly less extreme drought (SPEI of −1.9 in 2022 versus −2.1 in 2018). The role of delayed and compound effects of multiple warm and dry preceding summers on these needs more detailed analysis in a future study. In the South (yellow) region, we find, per unit area, a smaller response in 2022 compared to the North (blue) region in 2018 (2.0 and 3.0 gC m⁻² month⁻¹, respectively), even though South experienced locally more extreme conditions (mean 3-month SPEI of −2.2 for both regions, but an average VPD of 13.6 and 5.1 hPa, respectively). We hypothesise that this signifies a higher drought tolerance of the southern European vegetation, which is likely to be better adapted to high mean temperatures and low moisture availability than the northern forests hit in 2018[33,34].

Finally, the East (red) region, which was impacted only in 2022, contributed most to the NEE anomaly in the drought-influenced area (76%, also see Fig. 2e and Supplementary Table S10). This is a result that is based on the SiB4 model calculations, made necessary by a lack of sufficient atmospheric observation sites in the East. A drought very similar to the 2022 event occurred in 2015[11] but its carbon impacts remained under-studied, likely due to the same lack of observations. This gap in our monitoring capacity, also quantified in Storm et al (2023)[35], limits our understanding of drought impacts on net carbon uptake across pan-European forests, and will hamper the desired independent verification of forest carbon sequestration across the EU.

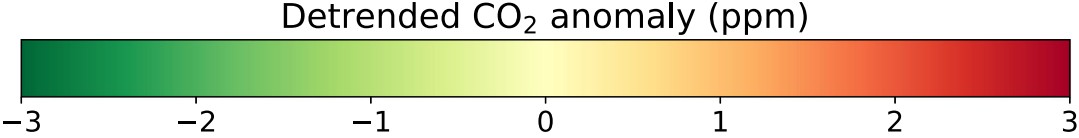

**Fig. 3 | The impact of the 2022 drought from an atmospheric CO₂ perspective.**
**a** Anomaly of atmospheric $CO_2$ for JJA, and **b** SON 2022 (see Methods and Supplementary Section C), relative to 2019–2021. Note that the influence of advection of background $CO_2$ is subtracted, leaving the surface flux influence on the anomaly (see Supplementary Table S3). **c** Shows (detrended) $CO_2$ monthly mean values in 2022 (red) and 2019–2021 (blue, with one standard deviation in grey) at the representative site Ochsenkopf (OXK). **d** Similar for Observatoire Pérenne de l'Environnement (OPE). OXK and OPE were selected to illustrate similar deviations found at other stations in the drought struck area. **e** Statistics (positive numbers mean higher atmospheric $CO_2$; center line, median; box limits, upper and lower quartiles; whiskers, 1.5x interquartile range; points, outliers) of the total atmospheric $CO_2$ anomalies (i.e. with background $CO_2$ variations, see Supplementary Section C) for all stations in the drought-affected area ($N = 20$) per season.

## Forest productivity

Reduced net carbon uptake in summer is a combination of reduced Gross Primary Productivity (GPP), partly balanced by reduced Terrestrial Ecosystem Respiration (TER)[36]. Reduced GPP results from a well-known mechanism to increase leaf-level water-use efficiency and reduce evaporative loss at the expense of carbon assimilation, detectable at leaf, ecosystem, and continental scale[37–39]. This in turn affects the canopy structure and leads to sub-optimal interception of

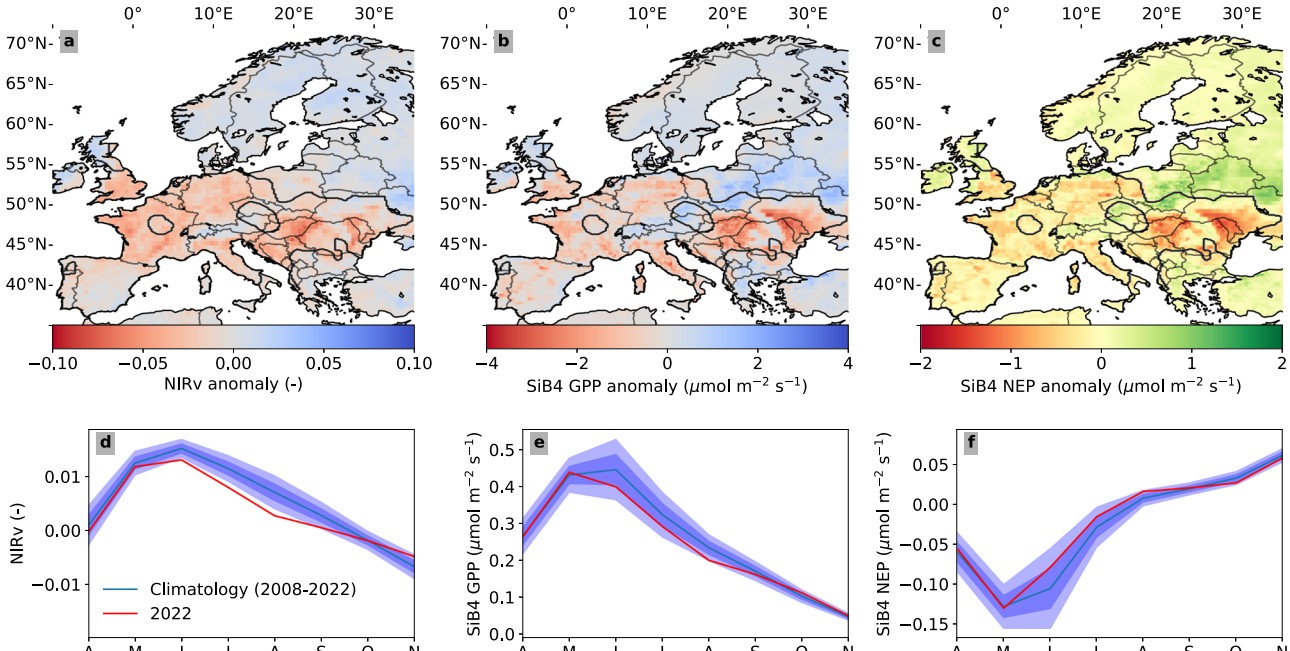

**Fig. 4 | Spatial patterns of reduced carbon uptake for 2022.** Spatial distribution of mean anomalies during the months June, July and August 2022 for **a** detrended near-infrared reflectance of vegetation (NIRv) calculated from MODIS surface reflectance, **b** gross primary production (GPP) simulated by the SiB4 biosphere model, and **c** Net Ecosystem Exchange (negative indicates less uptake) simulated by the SiB4 biosphere model. We coloured NEE differently to distinguish it from GPP and its proxy (NIRv). The climatology is based on 2008–2022. **d–f** Monthly mean values of the 2022 anomalies over the entire 2022 drought region (indicated by the black contours in (**a**–**c**) from April to November are shown underneath each corresponding map in red. The climatology and its one and two-sigma bandwidth are indicated in blue.

sunlight and widespread reductions in the reflection of near-infrared reflection by vegetation (NIRv[40]), which scales highly linearly with GPP[40]. For an extensive analysis of NIRv, see Supplementary Section G. This NIRv reduction is shown in Fig. 4 (see also Fig. 2g–j), and independently confirms large impacts on carbon uptake by vegetation across the southern, eastern and central region. The year 2022 ranks 1st or 2nd (behind 2018, depending on the gridcell) in magnitude over the 2000–2022 NIRv record (see Supplementary Fig. S14). Moreover, it highlights the eastern European region as a key impacted area with an unprecedentedly low summer NIRv on record (see also Supplementary Table S10). This partly results from the 2022 drought already starting in spring of 2022 in eastern Europe, propagating slowly towards the centre of the East area in Fig. 2, and further intensifying in June (see Supplementary Fig. S.13).

Following the approach we introduced in SM2020, we convert the anomaly in NIRv to GPP using its biome-specific linear relation to GPP derived from eddy-covariance observations (see Supplementary Material G). This results in averaged JJA reductions of $-44.1 \pm 17.4$ / $-50.7 \pm 18.5$ / $-47.3 \pm 18.2$ TgC/month over the Centre, South and Eastern region, respectively over the three summer months (total of $142.1 \pm 31.3$ TgC/month). Forests contributed $29.3 \pm 6.7$ TgC/month to this GPP anomaly ($-9.9 \pm 4.1$ / $-10.1 \pm 3.8$ / $-9.2 \pm 3.7$), corroborating the better resilience to drought than European grasslands and croplands found previously by Teuling et al. (2010)[41]. Independently, SiB4 calculates GPP-anomaly patterns highly similar to the observed NIRv (spatial correlation of $R = 0.78$, $N = 41$ bins, $p = 10^{-9}$, see Supplementary material D.1), integrating to $-12/-18/-25$ TgC/month for the same areas.

We consider the SiB4 estimate of the total GPP anomaly ($-55$ TgC/ month) a lower limit, based on its lower agreement with eddy-covariance (EC)-derived GPP across EC-sites ($R = 0.54$, $N = 14$). This is partly driven by differences in the environmental drivers, with at many sites the SPEI of the (gridded) ERA5 driver data substantially lower than in the EC-observations. As a consequence, SiB4 is unable to simulate

the observed near shutdown of photosynthesis during the most extreme period (see Fig. S6). But SiB4 captures the slope of $\Delta$GPP/ $\Delta$SPEI very well (SiB4: slope 2.0 vs EC: slope 2.5) which together with its high correlation with NIRV anomalies spatially gives credence to its drought response at larger scales. Note that we consider the NIRv-based GPP estimate an upper limit of the GPP reduction, as it is based on slopes derived at site level where the 2022 response was especially severe with near shutdowns at some sites, not seen in 2018 (also see SM2020 and Supplementary Section D.1).

### Vapour pressure deficit vs soil moisture

Our SiB4 results indicate that large atmospheric vapour pressure deficits are the dominant cause of reduced GPP in the southern and central regions in July-August 2022, while soil moisture deficits underlie the strongest impacts in eastern Europe. This is illustrated in Fig. 5, which shows the relative importance of three vegetation stress factors, calculated as separate factors that reduce assimilation and/or conductance (excess leaf temperature, high VPD in the leaf environment, and a deficit in root-zone soil moisture, (also see refs. [42–44]) for July and August of 2022, along with those of 2018. Soil moisture deficits, responsible for the most intense impacts on GPP in 2018, have played a smaller role in the central region in 2022 and were not the dominant driver of GPP reductions in southern France and Italy, where impacts were nevertheless high in 2022.

The important role of VPD in the southern and central regions is independently confirmed by local EC-observations, but soil moisture observations of sufficient quality and continuity were not available. At sites that locally experienced drought conditions (July 2022 SPEI < -1), GPP was reduced by 21% (also see Table 1). Furthermore, Fig. 5 indicates the extreme GPP anomaly for 2022 for the selected sites is mainly in the South region. Although GPP and TER are separate processes with independent responses, they often co-vary as they are sensitive to similar drivers and reduced GPP nearly always coincides with a reduction in ecosystem respiration[22,31,45–48]. These countering effects

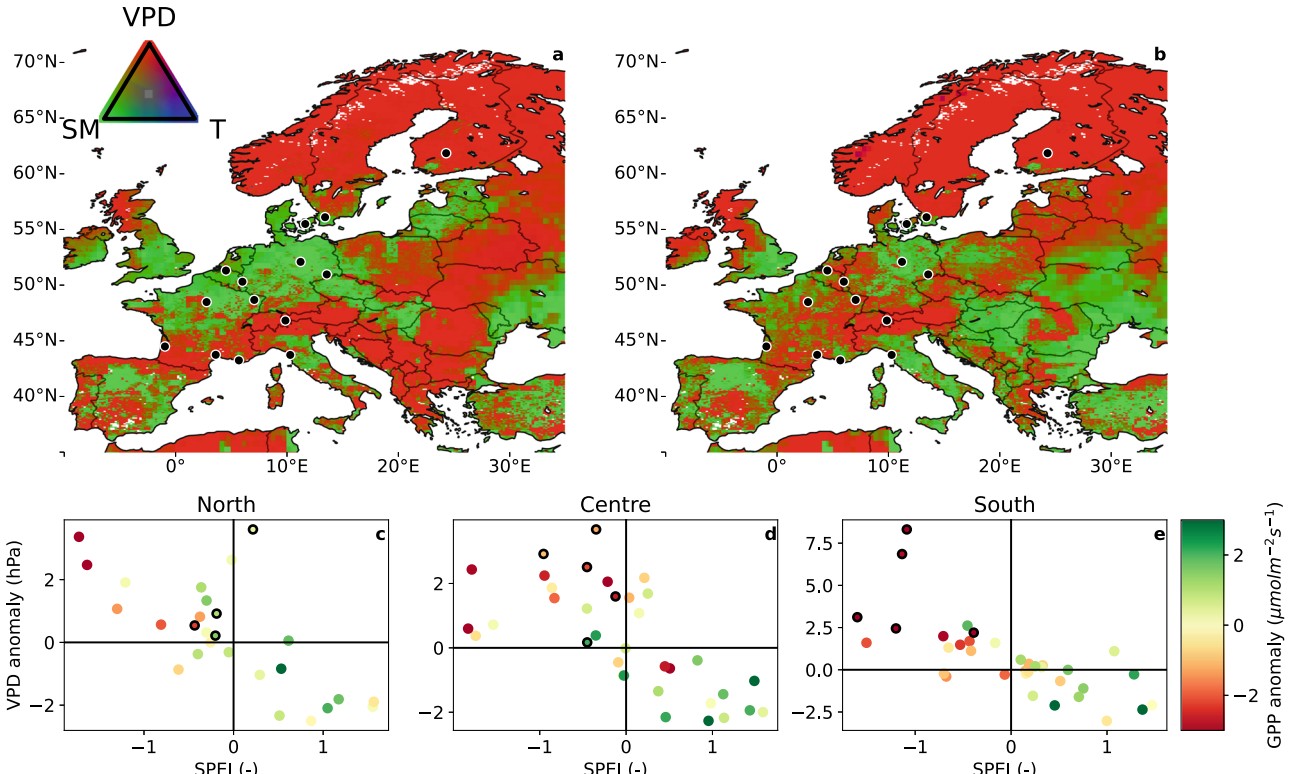

**Fig. 5 | Drivers of drought stress across Europe.** Dominant stress factors (VPD: atmospheric drought; SM: Soil moisture; T: canopy temperature) for July and August 2018 (**a**) and 2022 (**b**) in the SiB4 model. Note that the T stress is only the thermal stress at leaf-level, and VPD stress is only the atmospheric humidity component, without leaf-temperature effect. See Supplementary Section D3 for more information. The EC sites used are indicated by black dots. Bottom row: 3-month SPEI, averaged over July and August and VPD anomalies at available EC sites in the North (**c**), Centre (**d**) and South (**e**) regions (Supplementary Table S8) for the years 2016–2022, with 2022 marked with a black contour. Colours indicate measured GPP anomalies with respect to 2016-2022. A colourblind-friendly version of panels a and b is shown in Fig. S19.

**Table 1 | Meteorological and flux anomalies at forest EC sites during summer (JJA) and autumn (SON) of 2022, relative to 2019–2021 for gross primary production (GPP), net ecosystem exchange (NEE, positive denotes reduced uptake of carbon by the ecosystem), total ecosystem respiration (TER), Standardised precipitation and evaporation index (SPEI) and vapour pressure deficit (VPD)**

| | GPP ($\mu$ mol m$^{-2}$ s$^{-1}$) | | NEE ($\mu$ mol m$^{-2}$ s$^{-1}$) | | TER ($\mu$ mol m$^{-2}$ s$^{-1}$) | | SPEI (-) | | VPD (hPa) | |
|---|---|---|---|---|---|---|---|---|---|---|
| | JJA | SON | JJA | SON | JJA | SON | JJA | SON | JJA | SON |
| **Station name** | | | | | | | | | | |
| DK-Sor | −3.32 | 1.26 | 1.98 | −1.15 | −1.38 | 0.09 | −0.43 | −1.28 | 1.15 | 0.62 |
| IT-SR2 | −3.98 | −3.48 | 2.52 | 1.29 | −1.57 | −2.13 | −0.39 | −0.77 | 2.23 | 1.26 |
| FI-Hyy | 1.52 | 0.62 | −0.96 | −0.51 | 0.56 | 0.11 | −0.20 | −0.69 | 0.61 | -0.01 |
| BE-Bra | 1.43 | - | −1.38 | - | 0.06 | - | −0.45 | - | 0.59 | - |
| FR-FBn | −2.07 | 2.59 | 1.94 | −1.93 | −0.13 | 0.67 | −1.20 | 0.07 | 1.98 | 0.08 |
| FR-Pue | −2.39 | 1.83 | 1.63 | −0.84 | −0.76 | 0.99 | −1.14 | 0.17 | 6.80 | −1.15 |
| FR-Fon | 0.69 | 2.93 | −0.41 | −1.96 | 0.29 | 0.97 | 0.22 | 0.43 | 3.87 | −0.04 |
| FR-Hes | −4.08 | 4.48 | 2.47 | −3.31 | −1.61 | 1.15 | −1.60 | −0.61 | 3.43 | 0.49 |
| SE-Htm | 0.12 | 1.49 | −0.39 | −0.73 | −0.23 | 0.79 | −0.19 | −1.46 | 1.61 | 0.39 |
| DE-Tha | −1.78 | 0.60 | 1.50 | −0.46 | −0.28 | 0.13 | −0.35 | 0.77 | 4.13 | 1.93 |
| CH-Dav | −2.60 | −0.30 | 0.54 | −0.08 | −1.86 | −0.26 | −0.12 | −0.69 | 1.74 | 0.81 |
| BE-Vie | −1.09 | −0.08 | 0.94 | 1.12 | −0.20 | 1.04 | −0.96 | −0.56 | 3.30 | 0.03 |
| FR-Bil | −7.27 | −2.10 | 5.28 | 1.67 | −1.97 | −0.43 | −1.09 | −0.31 | 8.38 | 1.69 |
| DE-HoH | −2.70 | −0.41 | 1.71 | −0.31 | −0.72 | −0.29 | −0.45 | −0.17 | 2.98 | 0.88 |

reduce the impact of the drought on the net ecosystem exchange. Nevertheless, some sites that were most severely struck by the drought became a net source of CO$_2$ in July (FR-Pue) and averaged over July and August (FR-Bil) of 2022, even during daytime. This switch from sink to source of carbon during summer has not been observed before, and

proves that extreme conditions can cause carbon loss to the atmosphere even during the growing season. Moreover, if heat and atmospheric moisture demand are the main drivers, the extremes that drove the 2022 reversal will become part of normal climatological conditions as climate warming persists[20,49].

EC-observations that could confirm the dominance of soil moisture limitations calculated by SiB4 (Fig. 5) are lacking for the eastern region, but strong soil moisture depletion is simulated by ERA-Land[50] and observed through SMAP L-band satellite observations[51] of top-level soil moisture (Fig. S2). Depleted soil moisture is also very plausible given the early start of rainfall deficits in the eastern region. We see this as another example of the importance of spring-summer legacy effects in carbon uptake[31,52,53], but confirmation with local observations and more intense monitoring is needed to truly understand the driving mechanism.

## Warm autumn compensation

Beside the warm summer, 2022 also experienced the warmest autumn on record in Europe[3]. This warm autumn was accompanied by replenished soil water over large parts of the drought-affected area[3] (see also Supplementary Fig. S1). These conditions led to delayed leaf senescence, as seen by NIRv (Fig. 2 and Supplementary Figs. S13 and S15) and at EC sites (Table 1). The higher than normal NIRv is estimated to account for $66 \pm 12$ TgC/month higher GPP in October and November which is roughly 30% above normal. In SON, mean atmospheric $CO_2$ mole fractions at the stations in the area affected by the previous summer were 2.6 ppm lower (95% CI [2.4–2.9] ppm and standard deviation of 2.5 ppm) than normal (see also Fig. 3e), suggesting enhanced $CO_2$ uptake during the warm autumn and possibly a partial compensation for the reduced summer net carbon uptake. However, part of these low atmospheric $CO_2$ mole fractions are driven by advection of southerly air that was relatively low in $CO_2$, which reduced atmospheric $CO_2$ by 0.6 ppm in October (see Supplementary Table S3). This is also indicated by lower CO (Fig. S4) at atmospheric measurement stations.

We calculate that the enhanced uptake in October and November compensates up to 32% of the reduced uptake during the summer in the Centre and South region (see Table S11), much smaller than the 2018 warm spring compensation of ±75% found for 2018[31,52,53]. The lesser effect of a warm autumn compared to spring has also been found in deciduous forests[54], where better growing conditions in spring promote GPP more than TER, potentially due to the higher soil moisture and incoming radiation in spring[54]. Moreover, the mechanisms of enhanced autumn and spring uptake differ. Enhanced autumn uptake is controlled by late leaf senescence and continued photosynthesis in regions with sufficient light and heat[55,56], while enhanced spring carbon uptake is caused by early snow-melt and advanced accumulation of the temperature threshold for leaf-out, and plentiful sunlight[57,58]. In both mechanisms, accumulated impacts (i.e., higher than normal temperatures or incoming radiation due to lower cloud cover) affect phenology of vegetation. This is a difficult process to simulate mechanistically[55,58] as evidenced by the poorer performance of SiB4, as well as other vegetation models (see Supplementary Fig. S9) in autumn 2022. Note that we could not constrain this effect for the East region due to lack of measurements.

## Fires

Although in the Les Landes region in the south-west of France the drought spurred exceptionally large wildfires, the total loss of carbon through fires in Europe in 2022 is close to normal (14.6–16.0 TgC, see Supplementary information section I). Active fire counts from the Visible Infrared Imaging Radiometer Suite (VIIRS)[59] for the year 2022 show a positive anomaly relative to the previous 20 years, ranking fourth with total detections until August (Fig. S18). They similarly show a high anomaly in France, equivalent in magnitude to 2003 that had a similar water deficit and peak temperature. Importantly, in 2003 the fire anomaly occurred in Mediterranean areas whereas it took place in temperate areas during 2022 (Fig. S16). This resulted in more biomass burned per unit area in 2022 compared to 2003.

An assessment of these fires in France, based on Sentinel-2 observations of burned area at a 10m resolution and a 10 m map of impacted biomass density derived from Global Ecosystem Dynamics Investigation (GEDI) height and French National forest inventory (NFI) plot data was produced by Vallet et al.[60]. In their study they did not calculate emissions, but biomass lost from fires. Emissions of carbon gases and aerosols to the atmosphere should represent at most 50% of the biomass lost[60], given typical combustion completeness factors. Our best estimate of biomass loss derived from this study is of 0.5 Tg C y$^{-1}$. Specific to the fire season 2022 in France was that frequently burned Mediterranean forests and shrublands did not show an anomaly, but extreme fires occurred in regions where they have not been observed before (Atlantic pine forests in the South West, Brittany, Loire valley and Jura), affecting temperate forests with higher biomass, and thus leading to larger biomass loss rates. Although strongly affecting emissions from France, fires seem to have played a smaller role across the rest of Europe.

Despite high temperatures and extended drought, at European scale, the year 2022 only ranks 5th on record for Global Fire Assimilation System (GFAS) (2003-2022) carbon emissions[61]. It was characterised by below-normal emissions from March to June, followed by a fast rise in July, with a peak at 4.5 Tg C month$^{-1}$, which shows the highest July fire rate observed (Fig. S17). At country scale, the largest fire flux was in Spain with emissions of 1.8 Tg C month$^{-1}$ in July, accounting for 40% of the European emissions in that month. The second largest fire emissions emissions were in Portugal, peaking in August at (0.8 Tg C month$^{-1}$), about equal to France in July (0.6 Tg C month$^{-1}$). The integrated additional loss of carbon to the atmosphere from fires in 2022 is 5.2 TgC yr$^{-1}$ over the drought regions identified in this study (Table S10). Compared to 2003, the fire emissions over the entirety of Europe are very similar (15.2 and 14.6 TgC yr$^{-1}$ in 2003 and 2022, respectively), with 2003 having its peak carbon loss from fires in August (see also Supplementary Fig. S17).

## Discussion

We find a 2022 summer reduction of net carbon uptake of 56-62 TgC over the drought-affected area, which is similar to the reduced uptake in the summer of 2018 (50-66 TgC). But contrary to the drought of 2018, we do not find a large offset of this reduced uptake outside of the growing season as found previously in refs. 31,52,53. The 2022 situation thus resembles more the 2003 drought in its impact on net carbon uptake, which was estimated to have caused a net loss in the range of 20–500 TgC/yr[22,62,63]. The central estimate of these studies is substantially larger than our model calculation for 2022 (40.4 TgC/yr reduced uptake) and our inverse estimate for summer+autumn (50.2 TgC reduced uptake for JJASON, note that September contributed to the summer anomaly and not to the autumn compensation that occurred in October and November), but the 2003 event also covered a much larger area (nearly 4 million km²). Integrated GPP anomalies (derived from NIRv and EC observations) over the drought-affected area over the growing season were almost 50% larger in 2022 than in 2018 (see Supplementary Section G.1), which in turn was found higher than in 2003[58]. This difference is seen most strongly through NDVI[64] and NIRv (also see SI Section G), but also at site-level (−1.7 and −2.0 μmol m$^{-2}$ s$^{-1}$ ($N=14$) for 2018 and 2022, and −1.4 μmol m$^{-2}$ s$^{-1}$ reported in Ciais et al. (2005)[22] for 2003. In 2022 fires played a role in carbon loss from forests similar to 2003 (9.4 TgC and 8.8 TgC in JJA, respectively), which is larger than in 2018 (3.9 TgC). Together, these droughts show a substantial response of the European forest sink of $CO_2$ to drought, with the absence of favourable spring conditions, intensity of atmospheric heat, and record moisture deficits in summer as exacerbating factors.

Recent studies of tree-level responses to the 2018 drought have also shown this vulnerability of forest ecosystems to droughts[65]. Especially beech forests were found to be vulnerable[66,67]. Many of these

beech forests reside in the East region[68], which is a poorly monitored part of Europe[35]. With beech forests projected to have a growth reduction up to 90% by 2090 in areas that are projected to experience more severe droughts[67], it is imperative to better observe these forests.

Forest vulnerability to drought was also found in Haberstroh et al., (2022)[69], who found 47% of their 368 Scots pine trees to have died in 2020, due to legacy after the 2018 drought. Also Senf et al. (2021)[70] found reduced resilience due to drought, indicating persistent effects of drought. In this first assessment of the drought of 2022, we could not account for such legacy effects, that will likely play out in the next few years[71]. Therefore, these legacy effects could potentially aggravate the effects of the 2022 drought. These legacy effects should therefore be further explored in future studies, as should potential compound effects[53,72], such as the warm winter of 2022–2023.

The GPP impact on atmospheric $CO_2$ is typically larger and more variable than that of TER in summer[73,74]. Our study nevertheless suggests also a role of TER and soil moisture in the drought response, as we find a strong reduction of GPP which, according to the atmospheric $CO_2$ mole fraction constraints, does not fully balance observed NEE reductions without also considering a TER reduction. However, contrary to GPP, which can be estimated from satellite products, TER cannot currently be quantified on a large scale (although recently advances in the quantification of the temperature sensitivity of TER have been made[48]). Locally, our EC observations indeed show reduced TER and reduced GPP due to the drought in the South region in 2022. In the Centre region however, this effect is weaker (Supplementary Fig. S12). Although we lack the observations to verify this, we hypothesise the GPP response is dominated by the atmospheric stress through VPD, while the soil moisture limitation that would also affect TER is smaller. Biosphere models vary strongly in simulations of such a TER response[39,44,75] and recent work suggests a general overestimate of temperature sensitivity[48]. The good correspondence to atmospheric data nevertheless indicates a good NEE drought response in SiB4 which, together with a reasonable response in GPP, also suggests a reasonable TER response of SiB4.

Due to a lack of direct observations, our current results in the eastern region are based mostly on the biosphere model SiB4[44], downscaled to a higher resolution[32]. SiB4 was found to simulate the NEE response to droughts well in central and northern Europe[31,32], and also in this work its calculations agree well with observed atmospheric and EC-based anomalies where available (see Supplementary Figs. S6, S10). Nevertheless, the lack of $CO_2$ observing capacity, both in atmospheric $CO_2$ and ecosystem exchange over eastern European forests, as also indicated by others[35,76], remains worrisome. Especially eastern European forests could become an important part of the EU's goal to net-zero greenhouse gas emissions[77] but remain understudied until more extensive monitoring is realised. Most importantly, the impact of more frequent events like 2018 and 2022 need to be factored into potential carbon sequestration calculations by these, and other European forests[78].

This is especially relevant since hot extremes are projected to occur more over the entirety of Europe[14], with 75% of events attributable to global warming and the most extreme events most likely caused by anthropogenic influences. Indeed, attribution studies have found that the 2022 soil moisture drought is about 5 times as likely in 2022 as it was in pre-industrial climate[17]. In a 2-degree warmer climate, droughts are projected to occur once in every 10 years in Europe, instead of once per 100 years in pre-industrial climate[17]. This increased drought frequency has also been found by Spinoni et al. (2018)[18], who show that extreme droughts over Europe will be more frequent, with up to 1.2 events/decade more in the near future based on the intermediate RCP4.5 scenario. Although the main increase in expected drought events is in the South and West of Europe, also the poorly-observed but drought-sensitive eastern part of Europe is expected to experience more droughts[18],

Despite the mentioned limitations in the European carbon cycle monitoring, we also show in this study that with the build-up of ICOS, the infrastructure is present to analyse and quantify a major event in the European carbon cycle in near-real time. All atmospheric measurements, eddy-covariance measurements, satellite observations and model results we presented were available within a few days to maximally 3 months behind real-time. This is a stark improvement over the analysis of the 2018 drought that was concluded nearly 24 months after the event[28]. Especially, this offers good prospects for continuous integrated monitoring of the European carbon balance, aimed for by the EU's Copernicus program[79].

## Methods
### Selection of the regions
We identified the four different drought-affected regions in our analysis based on SPEI. We regridded the 3-month SPEI for July 2018 and 2022 from https://spei.csic.es/ (last access: 2023/01/26) to 0.2° by 0.1° using bi-linear interpolation. We selected continuous areas larger than 1.5 million km² with a SPEI < −1.2 (classified as severe drought), as visualised in Fig. 2.

Following this analysis, the area that was affected by drought in both 2022 and 2018 is called 'Centre'. The remainder of the area that was affected by the 2018 drought is called 'North'. The area that was only hit by the 2022 drought lies mainly in the southern and eastern part of Europe. We indicate the area that was affected by the 2022 drought, and under the high-pressure anomaly (see Fig. 1) as 'South', and the area away from the high-pressure anomaly as 'East'. The separation between South and East is based on country borders, with the East region covering Hungary, Romania, Bulgaria, Ukraine and Albania. For more information on the selected regions, see Supplementary material A.

### EC data
We used half hourly eddy-covariance and meteorological data at 14 forest sites. Data were downloaded from several ICOS datasets: WarmWinter2020[80], level 2 ICOS data for 2021 (and before when available) and NearRealTime (NRT) for 2022 (both from https://data.icos-cp.eu/, last access 17-04-2023). NRT data were cleaned and checked following Sabbatini et al. (2018)[81] and Pastorello et al. (2020)[82]. Subsequently, average fluxes were corrected for periods of low friction velocity (u*-threshold method), following Papale et al. (2006)[83]. All gaps were then filled using the marginal distribution sampling (MDS) algorithm from Reichstein et al. (2005)[84]. Finally, NEE partitioning into GPP and ecosystem respiration was done using the night-time based temperature response of NEE following Reichstein et al.[84]. u* calculation and filtering, gap-filling and fluxes partitioning were all performed using the REddyProc R package[85]. The sites are listed in Table S8 and an extensive analysis of the EC data is shown in Supplementary section F. Anomalies are calculated based on 2016-2021, excluding 2018. We note NEE from the atmospheric perspective, i.e. uptake by plants is negative and positive NEE indicates emissions into the atmosphere.

### Atmospheric data
Atmospheric mole fractions for the 26 used European stations for 2022 were taken from a pre-release of the ICOS 2023 Obspack (which is now available at[86]). These measurements were complemented with the Obspack 2022 release https://meta.icos-cp.eu/objects/2ESjwQy1qQRMEtcpiYPun2RO(last acces: 21/06/23). The stations are listed in Table S2. For each station, representative data was selected (i.e. night-time and well-mixed conditions for mountain stations and other stations, respectively). Based on data-availability, anomalies were calculated based on 2019–2021 after subtracting a station-specific linear trend fitted over yearly mean data from 2019 to 2021 from the data. For more information on the atmospheric

anomalies, as well as the inverse estimate, see Supplementary material C. Note that the inverse estimates derived from these data are referred to as Net Ecosystem Exchange (NEE) but they exclude the influence of large fires that we quantified separately (see SI Section I).

## NIRv data
We used NIRv calculated from the Bidirectional reflectance distribution function-corrected surface reflectances[40,87] at 0.5 by 0.5 degree from the Moderate Resolution Imaging Spectroradiometer (MODIS) satellite. To account for forest growth, we used linear detrended data, where the slope was calculated over the NIRv data from 2001-2021, which is the same period the anomalies were calculated over. Supplementary Material G contains a more in-depth description of the NIRv data used and the calculation of the GPP anomaly from NIRv data.

## SiB4 biosphere model
We used the SiB4 biosphere model[44] driven by ERA5 meteorology, with modified rooting zone depths to better account for soil moisture stress[31]. This product is downscaled to 0.1 x 0.2 ° based on land-use type following van der Woude et al (2023)[32]. Anomalies were calculated based on the data-availability, e.g., 2019–2021 for atmospheric comparisons, 2008–2021 for NIRv comparisons and 2016-2021 for EC-tower comparisons. For a more in-depth analysis of SiB4, see Supplementary material D. We refer to SiB4 flux calculations as Net Ecosystem Exchange (NEE) in the text, but we note that SiB4 calculates the GPP and TER fluxes between the vegetation and atmosphere and excludes fire emissions. This makes it comparable to the inverse estimates, and to SM2020, but deviates from the definitions of Ciais et al. (2022)[88] where NEE also includes fires and other fluxes.

## Data availability
MODIS reflectance data can be downloaded from https://modis.gsfc. nasa.gov/data/dataprod/mod13.php (last access 19/06/23) and GFAS data from https://ads.atmosphere.copernicus.eu/cdsapp#!/dataset/cams-global-fire-emissions-gfas?tab=overview (last access 19/06/23). Note that GFAS data for Europe after 2016, as well as CTE-HR NEE from SiB4 is available from https://doi.org/10.18160/2OZ1-AYJ2. Atmospheric $CO_2$ mole fractions used in this study are available at https://doi.org/10.18160/CEC4-CAGK and https://meta.icos-cp.eu/objects/2ESjwQy1qQRMEtcpiYPun2RO, (last access: 21/06/23) and EC measurements at https://doi.org/10.5281/zenodo.8056635.

## Code availability
SiB4 code is available on https://gitlab.com/kdhaynes/sib4v2_corral and CTE-HR code on https://git.wur.nl/ctdas/CTDAS.git. All code used for the analysis is available upon request.

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

## Acknowledgements

We acknowledge NWO/ENW (file number 2023.003) and SURF for supercomputing facilities. We furthermore acknowledge the PARIS project funded by the European Union under the Grant Agreement nr 101081430, NWO (Vidi grant VI.Vidi.213.143) and the Ruisdael Observatory, which is (partly) financed by the Dutch Research Council (NWO, grant number 184.034.015) for financial support. Furthermore, we thank the ICOS Carbon Portal for hosting the CTE-HR fluxes. We kindly acknowledge the provision of data from eddy-covariance sites[80], as well as from atmospheric sites[86]. The measurements at WAO are funded by the UK's National Centre for Atmospheric Sciences (NCAS). CO2 observations at Jungfraujoch are supported by ICOS Switzerland (ICOS-CH) Phase 3 (Swiss National Science Foundation, grant 20F120_198227). CH-Dav was supported by the SNF projects ICOS-CH Phase 1-3 (20FI21_148992, 143 20FI20_173691, 20F120_198227) and the EU project RINGO 730944. We acknowledge Guido van der Werf for his help on the fire flux estimates and kindly thank Imme Benedict for providing the GPH data. We also thank Thomas Koch and Saqr Munassar for pre-processing the meteorological data for the atmospheric transport with STILT. A.B., P.C., and S.S. were supported by ESA Carbon-RO (4000140982/23/I-EF).

## Author contributions

W.P. designed the study together with A.B., S.S., I.L., and P.C.; Avd.W., E.J., Y.X., and G.K. did the analyses. W.P. and Avd.W. wrote the manuscript with help from all co-authors. S.L., B.L., P.H.C., D.L., M.R., T.K., A.J., and D.K. provided observations shown in the manuscript. S.B. ran the STILT transport model. R.K. provided CTE-HR output. A.B. and S.S. provided OCN and JULES output data. P.C. provided fire estimates. All authors gave textual feedback on the manuscript.

## Competing interests

The authors declare no competing interests.
