## [Peer Review File · Nature Communications]

Temperature extremes of 2022 reduced carbon uptake by forests in EuropeReviewer #1 (Remarks to the Author):

This manuscript evaluates the carbon cycle effects of the European 2022 heat/drought and compares these to the event of 2018 by using a combination of ground observation from EC towers, atmospheric inversion approaches, terrestrial biosphere models and remote sensing products. The main result is that the 2022 event reduced summer carbon uptake across Europe and that, unlike in 2018 and during previous events, these reductions were not fully compensated for by anomalies during the side seasons of spring and fall. Consequently, the effect on annual carbon uptake was more pronounced during the year 2022. The ground observations are focused on forests, while spatial analyses includes the European continent and regions as a whole. The manuscript is well prepared and written, with a substantial supplementary material providing additional details on the analyses. The subject and focus of this manuscript is well within the scope of the journal and of great interest to the wider research community. This study presents novel insights on a very recent extreme event across Europe, which are highly relevant for better understanding the forest carbon. Nonetheless, the manuscript currently has some limitations and I would suggest various improvements before it can be recommended for publication.

General Comments:

- The Methods are currently explained in the extensive Supplement only and I am wondering, if it might be helpful for the readers to have at least a summary of the most important aspects in the main manuscript, i.e. for understanding the regional terms and definition of anomalies.
- The comparison of the triangular stress factor space (see Figure 5) currently appears inconsistent: temperature itself seems irrelevant as factor AND is by large included in VPD already. Accordingly, it might be more consistent to only compare SM & VPD for the stress factors.
- The overall discussion of the results would benefit from also comparing the impact of the two events (i.e. 2018 and 2022) qualitatively to the previous anomalies of 2003 and 2018, particularly as these impacts are well reported in the literature and/or data from these should be available from authors in this large team.
- I am wondering, if the anomalies in terms of drought/heat of 2018+2022 could be compared to the projections to be expected for end of the century across Europe? This could help putting the anomaly years into perspective compared to a future climate. Additionally, it might even be illustrative to also add the years 2003 & 2015 in that comparison for the respective discussion of previous research results.

Specific Comments:

- Figure 1: please add units of soil moisture
- Figure 2: the relative sizes of the large central map panel vs. the small detailed data side panels currently appears not optimal to me. The data panels contain the most important information here and should somehow be displayed larger for actually seeing the details between the fluxes/seasons/years.
- Figure 3: similarly, like Figure 2, the relative sizes of the panels do not reflect the importance of content here for me. The bottom panels with example sites shows more quantitatively the differences compares to the baseline and could be displayed more pronounced compared to the maps. I am also wondering if the mean/median seasonal cycle across all sites might give a better overall comparison here?
- Figure 4: I find it rather confusing the NEP panel is displayed in different color's than NIRv & GPP. What about using the same color scheme consistently across all panels or is there some reasoning against it? If so, it might help readers to shortly point that out in the caption.
- Figure 5b (bottom row) would benefit from displaying zero lines for both VPD & SPEI for clearly differentiating between positive/negative anomalies.
- Supplement A: incomplete reference "(see Figure ??)", please add the respective figure
- Supplement C: ditto as above for "Section ??" please add the respective reference
- Supplement E: it is unclear to me why this study uses incoming solar radiation as a rather unusual measure to remove data with suppressed (not 'surpressed') turbulence, and even if so, why with such a low threshold of 0.1 W/m². Solar radiation is rather a common measure to differentiate between day- and nighttime data, though typically using thresholds of 5 or 10 W/m². Please note that suppressed turbulence does not only occur during nighttime.
- Supplement Figure S6: unclear mix-up of fluxes in mol-units versus standard deviations of the climatology, which do not have a respective axis? In addition, some x-axis site labels are displayed

incomplete

- Supplement Figure S7: some y-axis labels (left & right, superscript & negative numbers) are displayed incomplete
- Supplement Figure S5: I am rather skeptical that the absolute precipitation anomalies in mm/month are a useful comparison as some of these sites typically have very little precipitation, and others large amounts in general. Alternatively, what about using relative precipitation anomalies, e.g. the rather common measure 'percent of normal'?
- Supplement G: NEP vs. NEE are typically also different regarding their signs, one being a productivity (NEP) and the other (NEE) being a measure of exchange from the biosphere perspective. Both are commonly converted via $\cdot -1$ for comparison. It might be helpful for the readers to elaborate on that.
- Supplement G1: ditto as above for "Figures ???" please add the respective reference
- Supplement G2: ditto as above for "Table ???" please add the respective reference. In addition, it could be helpful for readers to shortly explain the concept of the Z-scores in Figure S11.
- Supplement Figure S12: as the points appear having little meaning without a label, I would suggest to consider adding labels with site IDs. I would also suggest displaying colored linear regression lines for 2018 & 2022 (with stats displayed as text) for quantitatively comparing both drought year anomalies.
- Supplement G3: ditto as above for "Figure ???" please add the respective reference
- Supplement Figures S16+S17: insufficient resolution as pixel structure is visible
- Supplement Figure S18: labelling of year appears too small, please consider increasing size

Reviewer #2 (Remarks to the Author):

van der Woude et al. present an interesting analysis of the carbon balance effects of the 2022 drought in Europe. The paper covers a lot of ground and brings together many lines of evidence to show that VPD, temperature and soil moisture anomalies are resulting in large perturbations to the potential for forest carbon sequestration. Many comparisons are made to previous droughts, particularly the 2018 drought, but the unique setting of the 2022 drought offers further insights into large scale continental effects of changing climate regimes. I admit to feeling that the evidence builds slowly and times seems a bit frustrating but the brief, strong discussion does a nice job of finally pulling the pieces together. With no major criticisms of the study I did find several small issues to clarify and found several of the figures in need of improvement.

Introduction

First paragraph – Replace "the continent" with "the European continent" or simply "Europe"

.

Anatomy of a Summer drought

First paragraph last sentence – Beech is a type of broadleaf forests?

Net Carbon Exchange Impacts

Figure 2 – this is a very busy graph that is very poorly described when introduced in the text and legend.

For example "(b) Fire fluxes, taken from GFAS28 over the South (blue) and North (red) regions" but the lines on the graph are blue and yellow rather than blue and red and the map and other panels use blue as north and yellow as the south?

The 'hatching' used in panels c-f is too widely spaced

In panels g-j I assume the gray line is the "climatology between 2016-2022" but this is never identified.

Figure 3 is similarly not well presented.

The legend states that "together with CO₂ monthly mean values in 2022 (red) and 2019-2021 (grey, with one standard deviation in blue)" yet it appears as though the values are in red and blue with the standard deviation of 2021 in grey.

What makes Ochsenkopf (OXK, C) and Observatoire pérenne de l'environnement (OPE, d)

representative?

Vapor pressure deficit vs. soil moisture

"This switch from sink to source of carbon sink during summer" – I think there is an extra sink.

Discussion

The discussion of TER is extremely interesting but the discussion becomes somewhat confusing when the following paragraph states that NEP can be successfully simulated – why are these simulations not affected by the same inaccuracies in TER?

Supplementary files

Figure S6 – the x-axis legend is cut off

1 Reviewer 1 (Anonymous)

1.1 General comments.

- *This manuscript evaluates the carbon cycle effects of the European 2022 heat/drought and compares these to the event of 2018 by using a combination of ground observation from EC towers, atmospheric inversion approaches, terrestrial biosphere models and remote sensing products. The main result is that the 2022 event reduced summer carbon uptake across Europe and that, unlike in 2018 and during previous events, these reductions were not fully compensated for by anomalies during the side seasons of spring and fall. Consequently, the effect on annual carbon uptake was more pronounced during the year 2022. The ground observations are focused on forests, while spatial analyses includes the European continent and regions as a whole.*

The manuscript is well prepared and written, with a substantial supplementary material providing additional details on the analyses. The subject and focus of this manuscript is well within the scope of the journal and of great interest to the wider research community. This study presents novel insights on a very recent extreme event across Europe, which are highly relevant for better understanding the forest carbon. Nonetheless, the manuscript currently has some limitations and I would suggest various improvements before it can be recommended for publication.

We thank the anonymous reviewer for their positive, critical, and constructive feedback. Below, we address the reviewer's suggestions one by one, with the reviewer comments in italic and our answers in blue. When we refer to a figure, we refer to the figure number of the original manuscript and when we refer to line numbers, we refer to line numbers in the revised tracked changes document.

1.2 Detailed comments.

- *The Methods are currently explained in the extensive Supplement only and I am wondering, if it might be helpful for the readers to have at least a summary of the most important aspects in the main manuscript, i.e. for understanding the regional terms and definition of anomalies.*

We agree with the reviewer and we have added a short methods section to the end of the main text (L.298) describing how we selected the different regions and describes the most important data used, and their processing.

- *The comparison of the triangular stress factor space (see Figure 5) currently appears inconsistent: temperature itself seems irrelevant as factor AND is by large included in VPD already. Accordingly, it might be more consistent to only compare SM & VPD for the stress factors.*

We understand that this figure may have raised some confusion with the reviewer, as vapour pressure deficit (VPD) indeed is partly a function of temperature (T), which would make their separation superfluous.

However, what we try to convey in this figure are three dominant and different pathways through which plant stress reduces photosynthesis. These are (a) the stress resulting from large evaporative demand in the atmosphere, which we refer to as "VPD stress", (b) stress resulting from a lack of plant available water in the soil ("root-zone SM stress"), and (c) thermal stress that impairs the enzymes in the leaves that play a role in photosynthesis ("T-stress", see e.g. Hu et al. [2020]). The latter is the right-hand side of the triangle. VPD stress in the SiB4 numerical model is separately calculated from leaf T-stress, based on atmospheric relative humidity through the Ball-Berry equation. In the study we show VPD instead of RH because it is more informative on absolute water potential at the leaf stomata, and we refer to the associated stress as "VPD-stress". We now explain the differences in more detail in the caption of the figure to help the reader understand the three axes of stress.

Figure 1: Stress factors for the three different stresses in SiB4 for the South of Europe in JJA 2022. The lower the stress factor, the lower GPP is. Only the most dominant (lowest) stress factor is applied to reduce modeled photosynthesis.

Another point that the reviewer rightfully mentions, is the lack of T stress shown in Figure 5. In SiB4, the photosynthetic capacity is calculated based on the most limiting stress. We have added a figure with the maximum T stress for JJA 2022, as well as the maximum SM and VPD stress (Fig. 1). For clarity, we zoomed in on the South of Europe, where temperature (and thus thermal stress) is highest. From this figure, it is clear that although the high temperatures give stress to the vegetation, it is not the most dominant stress, as also discussed in Fu et al. [2022], Liu et al. [2020]. We have added a note about this in the supplement in L.743..

- *The overall discussion of the results would benefit from also comparing the impact of the two events (i.e. 2028 and 2022) qualitatively to the previous anomalies of 2003 and 2018, particularly as these impacts are well reported in the literature and/or data from these should be available from authors in this large team.*

We agree with the reviewer that comparing these different droughts would improve our discussion. In the manuscript, we try to compare the 2022 drought to the 2018 drought, but indeed also 2003 is an interesting case study. Therefore, we have added additional comparisons to the 2003 drought to the discussion (L.234-247), in which we show that the 2003 drought showed a less strong response, even though first estimates of its NEP anomaly were considerably larger [Ciais et al., 2005]. Moreover, we have added a reference to the 2015 drought, of which the carbon impacts remain under-studied, in L.115.

- *I am wondering, if the anomalies in terms of drought/heat of 2018+2022 could be compared to the projections to be expected for end of the century across Europe? This could help putting the anomaly years into perspective compared to a future climate. Additionally, it might even be illustrative to also add the years 2003 & 2015 in that comparison for the respective discussion of previous research results.*

We agree with the reviewer that such an analysis of future projections would be valuable. However, placing one event in the perspective of climate change remains challenging. A possibility to do this would be e.g. by attribution studies. For example, Schumacher et al. [2023] found that the 2022 soil moisture drought is about 5 times as likely in 2022 as it was in pre-industrial times, owing to rising global temperatures. Moreover, they estimate the return time of such a drought to increase from once in every 100 years (pre industrial), via 1 in every 20 years (current) to 1 in every 10 years with 2 degrees warming. This increased drought frequency has also been found by Spinoni et al. [2018], who show that extreme droughts over Europe will be more frequent, with an increase of up to 1.2 events/decade in the near future, based on the intermediate RCP4.5 scenario.

Given these studies, we think a new attribution analysis would require a large, dedicated effort and is outside the scope of this research. To put the anomalies in perspective, as the reviewer suggested, we have added a paragraph to the discussion (L.284-291), which includes the points made above.

1.3 Technical corrections.

Specific Comments:

- *Figure 1: please add units of soil moisture*

We agree with the reviewer that adding this unit (m^3m^{-3} , which we simplified to unitless) improves the understanding of the figure, and have changed it.

- *Figure 2: the relative sizes of the large central map panel vs. the small detailed data side panels currently appears not optimal to me. The data panels contain the most important information here and should somehow be displayed larger for actually seeing the details between the fluxes/seasons/years.*

We have updated the relative sizes of the figures. Note that we might solicit assistance from the journals graphical design team to further improve the setting of this figure.

- *Figure 3: similarly, like Figure 2, the relative sizes of the panels do not reflect the importance of content here for me. The bottom panels with example sites shows more quantitatively the differences compares to the baseline and could be displayed more pronounced compared to the maps. I am also wondering if the mean/median seasonal cycle across all sites might give a better overall comparison here?*

We agree with the reviewer, and thank them for the suggestion to add a mean seasonal cycle across all sites. To inform the reader the statistics of the mole fraction anomalies, we have added boxplots of these anomalies as a subplot to the figure.

- *Figure 4: I find it rather confusing the NEP panel is displayed in different colors than NIRv & GPP. What about using the same color scheme consistently across all panels or is there some reasoning against it? If so, it might help readers to shortly point that out in the caption.*

We have chosen different colour maps for GPP+NIRv and NEP, as we wanted to indicate the difference between GPP (for which NIRv is a proxy) and NEP. We agree with the reviewer that this should be elaborated on in the caption, and have added it.

- *Figure 5b (bottom row) would benefit from displaying zero lines for both VPD & SPEI for clearly differentiating between positive/negative anomalies.*

We agree with the reviewer and have added these lines.

- *Supplement A: incomplete reference “(see Figure ??)”, please add the respective figure*

We thank the reviewer for noting this. In the updated version, we moved this section (based on the reviewers comment about including a methods section), and have made sure the correct reference is present.

- *Supplement C: ditto as above for “Section ??” please add the respective reference*

We thank the reviewer for noting this and have added the respective reference to the main text.

- *Supplement E: it is unclear to me why this study uses incoming solar radiation as a rather unusual measure to remove data with suppressed (not ‘surpressed’) turbulence, and even if so, why with such a low threshold of 0.1 W/m^2 . Solar radiation is rather a common measure to differentiate between day- and nighttime data, though typically using thresholds of 5 or 10 W/m^2 . Please note that suppressed turbulence does not only occur during nighttime.*

There was indeed a confusion in the writing of the EC methods, and we thank the reviewer for noting it. We have updated the text (in the newly added Methods section, L.310) to better explain the preprocessing pipeline. For completeness, we have added the relevant part of the methods paragraph here, with some additional explanation:

NRT data were cleaned and checked following Sabbatini et al. [2018], Pastorello et al. [2020]. Subsequently, average fluxes were corrected for periods of low friction velocity (u^* method), following Papale et al. [2006]. All gaps were then filled using the marginal distribution sampling (MDS) algorithm from Reichstein et al. [2005]. Finally, NEE partitioning into GPP and ecosystem respiration was done using the night-time based temperature response of NEE following Reichstein et al. [2005]. u^* calculation and filtering, gap-filling and fluxes partitioning were all performed using the REddyProc R package [Wutzler et al., 2018].

We note that the REddyProc package uses a threshold of 10 W/m^2 for separating day- and night-time data.

Apart from the pre-processing in the REddyProc software, we used an incoming radiation threshold of 0.1 Wm^{-2} to differentiate between night- and daytime data for the analyses shown. Following the suggestion of the reviewer, we have changed that to 10 Wm^{-2} , consistent with the REddyProc threshold. We note that this change leads to the exclusion of 7.5% of the data used, which affects the main text in L.162, decreasing the effect of the drought from 27% to 21%. Additionally, it affects Table 1 and Figure 5. Moreover, it affects Figures S5, S6, S11 and S12 (in the updated manuscript) and Tables S4, S5 and S9. We note that differences do not affect the main findings of the manuscript.

We also learned that an error occurred in the (external) processing of precipitation data of the site CH-Dav. We have updated our plots and tables with the new, improved data.

Moreover, we noted that we made a mistake in calculating the SPEI for the different sites. Instead of using potential evapotranspiration, we used the actual evapotranspiration. In the new version of the manuscript, this is updated and we have updated the text to represent the calculation of the potential evapotranspiration. We note that the differences in resulting SPEI are small, often smaller than 0.3, and that this does not affect the main results.

- *Supplement Figure S6: unclear mix-up of fluxes in mol-units versus standard deviations of the climatology, which do not have a respective axis? In addition, some x-axis site labels are displayed incomplete*

The indicated standard deviation of the climatological fluxes, as well as the anomalies, are already in flux-units ($\mu\text{mol m}^{-2} \text{ s}^{-1}$). However, we note that in the PDF rendering, some labels were inadvertently cut off, which we have fixed in the revised version.

- *Supplement Figure S7: some y-axis labels (left & right, superscript & negative numbers) are displayed incomplete*

We thank the reviewer for noting this and have corrected this.

- *Supplement Figure S5: I am rather skeptical that the absolute precipitation anomalies in mm/month are a useful comparison as some of these sites typically have very little precipitation, and others large amounts in general. Alternatively, what about using relative precipitation anomalies, e.g. the rather common measure ‘percent of normal’?*

We assume here that the reviewer meant Table S5, instead of Figure S5.

We agree with the reviewer that displaying precipitation anomalies in % is more intuitive, and have changed the table. We did this as well for Table S4.

- *Supplement G: NEP vs. NEE are typically also different regarding their signs, one being a productivity (NEP) and the other (NEE) being a measure of exchange from the biosphere perspective. Both are commonly converted via '-1 for comparison. It might be helpful for the readers to elaborate on that.*

We thank the reviewer for noting this, and helping clarifying the paper for the general public. The definition of our fluxes gave rise to discussion within our author team as well, since the recently proposed definitions by Ciais et al. [2022] do not fit perfectly with the two fluxes we present in the main text. (1) the SiB4 biosphere model fluxes most strongly represent the difference between GPP and TER, and thus "NEP" would fit, although the respiration of SiB4 represents all carbon losses, including for example reduced species like VOCs. (2) For the inverse estimates, NEE would fit best as it represents all exchange from the surface, except that we excluded most of the fire impacts by either not being sensitive to them, or by filtering large peaks in the data before inverting. As a result, both fluxes represent a much similar anomaly: that of the carbon uptake by vegetation, which is also measured by EC sites. Since we refer to this as "NEE" in Smith et al. [2020] -like was done in many inverse modeling papers previously- we decided here to refer to all our fluxes as NEE, with the specific note (L.345) that we quantify fire fluxes separately. Throughout the paper, we are careful to explain the reader what flux component we refer to (see L.321 and L.698 for examples). We also specifically address the sign convention in the text now (in L.81, as well as in the caption of Table 1 and Figures 2 and 4.)

- *Supplement G1: ditto as above for "Figures ???" please add the respective reference*

We thank the reviewer for noting this and have added the reference.

- *Supplement G2: ditto as above for "Table ??" please add the respective reference. In addition, it could be helpful for readers to shortly explain the concept of the Z-scores in Figure S11.*

We have added the reference, as well as an explanation of the calculation of the Z-score to Figs. S7, S8 and S12 in the new document, and thank the reviewer again for helping clarifying the manuscript for the broader public.

- *Supplement Figure S12: as the points appear having little meaning without a label, I would suggest to consider adding labels with site IDs. I would also suggest displaying colored linear regression lines for 2018 & 2022 (with stats displayed as text) for quantitatively comparing both drought year anomalies.*

We thank the reviewer for bringing up this point, which made us reconsider this figure and improve it according to the suggestion. We moreover expanded the panels to show extra details as below.

The original figure mostly showed that SiB4 does not capture the site-to-site variations of GPP anomalies as well as it seems to do for NIRv anomalies. This is also reflected in the lower correlation coefficient ($R=0.54$, $N=14$) for the 2022 data we plotted in Figure S12, and provided in the Table below. The reason for this is two-fold: firstly, at EC-site level the correspondence of the environmental drivers at grid level (ERA5) compared to site level is much worse in 2022, compared to 2018, with at many sites the SPEI of the ERA5 driver data substantially lower than in the EC-observations (the mean difference of SPEI in July smaller than 0.5 only for 3 out of 14 sites). This was less so in 2018 when the correlation coefficients between site-observed and SiB4-modelled GPP and NEP were higher, and the slope of the site-to-site difference steeper suggesting we captured more of the variation across space. Secondly, SiB4 is unable to simulate the observed near shutdown of photosynthesis during the most extreme period of 2022, thus missing the high end of the impacted sites.

However, this does not mean that SiB4 has a poor drought response in GPP, it just does not do well representing each site with the gridded driver data provided, a well-known shortcoming in evaluations of biosphere models. So in addition, we have created a plot where the SPEI anomaly of model and EC observations is on the x-axis, and we regress it against the corresponding GPP-anomalies. This is shown in an expanded version of Figure S12, with

regression coefficients given in Table 2. It shows that SiB4 captures the *slope* of $\Delta\text{GPP}/\Delta\text{SPEI}$ very well (SiB4: slope 2.0 vs EC: slope 2.5 ($\mu\text{ mol m}^{-2}\text{s}^{-1}$) and thus the drought *response* across the SPEI gradient. This, together with its high correlation with NIRV anomalies spatially, lends credence to its drought response at larger scales. Note that because of the offset in the fitted ΔGPP , and the lack of "shutdown" at extreme SPEI's, we now explain the SiB4 GPP integrals as a lower limit to the estimates in the main text (L.142-150), as well as in the Supplement (L. 721-736). We show the updated Figure S12 (now S13) here for completeness (2). As we agree with the reviewer on its points regarding the statistics (Table 1), we have added these. As the statistics shown in Table 1 already capture the original scatterplots in the Figure, we have removed these. Note that for the ΔGPP -SPEI plot, we opted not to add labels, as the main point is not to show the magnitude of the reduction in GPP at sites, but the relation between SPEI and GPP reduction.

Table 1: Linear regression statistics of ΔGPP from EC sites and SiB4 in corresponding gridcells.

	Slope	Intercept	R	P
GPP 2018	0.56	0.32	0.86	7E-5
NEP 2018	0.51	0.20	0.81	4E-4
GPP 2022	0.25	0.30	0.54	0.047
NEP 2022	0.24	0.11	0.59	0.027

Table 2: Linear regression statistics for SiB4 and EC-measured GPP and NEP

	Slope	Intercept	R	P
EC	2.5	-0.46	0.53	0.049
SiB	2.0	0.77	0.81	5E-4

- *Supplement G3: ditto as above for "Figure ???" please add the respective reference*

We have added the reference.

- *Supplement Figures S16+S17: insufficient resolution as pixel structure is visible*

We agree with the reviewer and have increased the resolutions

- *Supplement Figure S18: labelling of year appears too small, please consider increasing size*

We agree with the reviewer and have increased the label size.

Figure 2: a) GPP and anomalies in JJA of 2018 (blue) and 2022 (orange) at SiB4 and EC sites; b) like a) but for NEE; a) GPP anomalies in JJA 2022 as function of SPEI for EC (blue) and SiB4 (orange); b) Seasonal cycle of the site FR-Hes. The solid blue (orange) line shows the measured (simulated) GPP in 2022 and the blue (orange) shading the climatology (2016-2021, mean $\pm 1\sigma$). We note that the SPEI in July of 2022 based on site-measured meteorology is -1.7, and the SPEI according to SiB4 driver data is -0.16.

2 Reviewer 2 (anonymous)

van der Woude et al. present an interesting analysis of the carbon balance effects of the 2022 drought in Europe. The paper covers a lot of ground and brings together many lines of evidence to show that VPD, temperature and soil moisture anomalies are resulting in large perturbations to the potential for forest carbon sequestration. Many comparisons are made to previous droughts, particularly the 2018 drought, but the unique setting of the 2022 drought offers further insights into large scale continental effects of changing climate regimes. I admit to feeling that the evidence builds slowly and times seems a bit frustrating but the brief, strong discussion does a nice job of finally pulling the pieces together. With no major criticisms of the study I did find several small issues to clarify and found several of the figures in need of improvement.

We thank the anonymous reviewer for their feedback and constructive comments, which help to improve the paper. Below, we address the comments one by one, with the reviewer comments in italic and our answers in blue. When we refer to line numbers, we refer to line numbers in the revised tracked changes document.

2.1 Detailed comments:

- *Introduction First paragraph – Replace “the continent” with “the European continent” or simply “Europe”*

We have changed the sentence in L.31.

- *Anatomy of a Summer drought First paragraph last sentence – Beech is a type of broadleaf forests?*

We thank the reviewer for noting this, and have changed the text to only contain beech forests in L.66.

- *Net Carbon Exchange Impacts Figure 2 – this is a very busy graph that is very poorly described when introduced in the text and legend. For example “(b) Fire fluxes, taken from GFAS28 over the South (blue) and North (red) regions” but the lines on the graph are blue and yellow rather than blue and red and the map and other panels use blue as north and yellow as the south? The ‘hatching’ used in panels c-f is too widely spaced In panels g-j I assume the gray line is the “climatology between 2016-2022” but this is never identified.*

We thank the reviewer for their comments on this important figure and helping improving it. We have tried to address the concern but we find it not easy to make a more aesthetically pleasing figure. We are considering to solicit assistance from the journals graphical design team to improve the setting of this figure.

- *Figure 3 is similarly not well presented. The legend states that “together with CO2 monthly mean values in 2022 (red) and 2019-2021 (grey, with one standard deviation in blue)” yet it appears as though the values are in red and blue with the standard deviation of 2021 in grey. What makes Ochsenkopf (OXK, C) and Observatoire pérenne de l’environnement (OPE, d) representative?*

We thank the reviewer for noting the confusion in the caption and have changed it.

We have chosen OXK and OPE, because they show anomaly values that correspond well to other stations shown on the maps in the same figure (we have added a mean seasonal deviation to the subplot, which corresponds well to these two stations). We have added a short explanation about this in the caption as well.

- *Vapor pressure deficit vs. soil moisture “This switch from sink to source of carbon sink during summer” – I think there is an extra sink.*

We thank the reviewer for noting this textual inconsistency and have removed the second ‘sink’ in L.169.

- *Discussion* The discussion of TER is extremely interesting but the discussion becomes somewhat confusing when the following paragraph states that NEP can be successfully simulated – why are these simulations not affected by the same inaccuracies in TER?

The reviewer raises a good point here, that indeed deserves further explanation in the text.

Many previous studies [Baldocchi, 2008, Lasslop et al., 2010, le Maire et al., 2010] have shown that, although GPP and TER are separate gross fluxes resulting from different processes with individual responses, they often co-vary as they are sensitive to similar drivers. It is well-known that TER has a (soil-)T dependence, but also low soil moisture is known to inhibit TER [Orchard and Cook, 1983]. The variations in GPP are often the dominant variation in NEE, especially during daytime in summer, when GPP can be twice as large as TER [Tolk et al., 2009]. The GPP response is also relatively well-observed in EC-measurements, and in satellite observations such as NIRv. Therefore, GPP is the component of SiB4 modeling that we have been able to validate better than TER. The TER drought response in itself is partly observable at EC-sites and presented in Table 1, confirming that it reduces during the drought just like GPP, and as explained above. The balance of the GPP and TER change constitutes the simulated NEE response, which is independently constrained by the observed changes in atmospheric CO₂ mole fractions that it drives. Multiple combinations of GPP and TER can lead to the same NEE, and there is also a degree of cancelling errors in the gross fluxes, and therefore a good NEE response is easier to obtain than good responses in gross fluxes (see Peters et al (2018) and the multi model-data comparisons in its Supplement). For SiB4, we find that from a reasonable simulation of GPP, we can infer that the TER response is credible at least in sign and magnitude, and its NEE response is represented well, despite flaws in gross fluxes.

In the discussion (L.258-275) of the revised manuscript, we have now explicitly explained the indirect and weaker constraints on TER offered this way, in an attempt to address the reviewer’s valid comment. The new paragraph reads:

The GPP impact on atmospheric CO₂ is typically larger and more variable than that of TER in summer Tolk et al. [2009], Piao et al. [2020]. Our study nevertheless suggests also a role of TER and soil moisture in the drought response, as we find a strong reduction of GPP which, according to the atmospheric CO₂ mole fraction constraints, does not fully balance observed NEE reductions without also considering a TER reduction. However, contrary to GPP, which can be estimated from satellite products, TER cannot currently be quantified on a large scale (although recently advances in the quantification of the temperature sensitivity of TER have been made Sun et al. [2023])). Locally, our EC observations indeed show reduced TER and reduced GPP due to the drought in the South region in 2022. In the Centre region however, this effect is weaker (Supplementary Fig. S12 of the revised manuscript). Although we lack the observations to verify this, we hypothesise the GPP response is dominated by the atmospheric stress through VPD, while the soil moisture limitation that would also affect TER is smaller. Biosphere models vary strongly in simulations of such a TER response Clark et al. [2011], Peters et al. [2018], Haynes et al. [2020] and recent work suggests a general overestimate of temperature sensitivity Sun et al. [2023]. The good correspondence to atmospheric data nevertheless indicates a good NEE drought response in SiB4 which, together with a reasonable response in GPP, also suggests a reasonable TER response of SiB4.

- *Supplementary files Figure S6 – the x-axis legend is cut off*

We have resolved this issue.

References

- D. Baldocchi. 'Breathing' of the terrestrial biosphere: lessons learned from a global network of carbon dioxide flux measurement systems. *Australian Journal of Botany*, 56(1):1, 2008. ISSN 0067-1924. doi: 10.1071/BT07151. URL www.publish.csiro.au/journals/ajbhttp://www.publish.csiro.au/?paper=BT07151.
- P. Ciais, M. Reichstein, N. Viovy, A. Granier, J. Ogée, V. Allard, M. Aubinet, N. Buchmann, C. Bernhofer, A. Carrara, F. Chevallier, N. De Noblet, A. D. Friend, P. Friedlingstein, T. Grünwald, B. Heinesch, P. Keronen, A. Knohl, G. Krinner, D. Loustau, G. Manca, G. Matteucci, F. Miglietta, J. M. Ourcival, D. Papale, K. Pilegaard, S. Rambal, G. Seufert, J. F. Soussana, M. J. Sanz, E. D. Schulze, T. Vesala, and R. Valentini. Europe-wide reduction in primary productivity caused by the heat and drought in 2003. *Nature*, 437(7058):529–533, 9 2005. ISSN 0028-0836. doi: 10.1038/nature03972. URL <https://www.nature.com/articles/nature03972>.
- P. Ciais, A. Bastos, F. Chevallier, R. Lauerwald, B. Poulter, J. G. Canadell, G. Hugelius, R. B. Jackson, A. Jain, M. Jones, M. Kondo, I. T. Lujckx, P. K. Patra, W. Peters, J. Pongratz, A. M. R. Petrescu, S. Piao, C. Qiu, C. Von Randow, P. Regnier, M. Saunois, R. Scholes, A. Shvidenko, H. Tian, H. Yang, X. Wang, and B. Zheng. Definitions and methods to estimate regional land carbon fluxes for the second phase of the REgional Carbon Cycle Assessment and Processes Project (RECCAP-2). *Geoscientific Model Development*, 15(3):1289–1316, 2 2022. ISSN 1991-9603. doi: 10.5194/gmd-15-1289-2022. URL <https://gmd.copernicus.org/articles/15/1289/2022/>.
- D. B. Clark, L. M. Mercado, S. Sitch, C. D. Jones, N. Gedney, M. J. Best, M. Pryor, G. G. Rooney, R. L. H. Essery, E. Blyth, O. Boucher, R. J. Harding, C. Huntingford, and P. M. Cox. The Joint UK Land Environment Simulator (JULES), model description – Part 2: Carbon fluxes and vegetation dynamics. *Geoscientific Model Development*, 4(3):701–722, 9 2011. ISSN 1991-9603. doi: 10.5194/gmd-4-701-2011. URL <https://gmd.copernicus.org/articles/4/701/2011/>.
- Z. Fu, P. Ciais, I. C. Prentice, P. Gentile, D. Makowski, A. Bastos, X. Luo, J. K. Green, P. C. Stoy, H. Yang, and T. Hajima. Atmospheric dryness reduces photosynthesis along a large range of soil water deficits. *Nature Communications*, 13(1):989, 2 2022. ISSN 2041-1723. doi: 10.1038/s41467-022-28652-7. URL <https://doi.org/10.1038/s41467-022-28652-7https://www.nature.com/articles/s41467-022-28652-7>.
- K. Haynes, I. T. Baker, and A. S. Denning. The Simple Biosphere Model, Version 4.2: SiB4 Technical description. *Colorado State University*, (February), 2020. URL <https://hdl.handle.net/10217/200691>.
- X. M. Hu, S. Crowell, Q. Wang, Y. Zhang, K. J. Davis, M. Xue, X. Xiao, B. Moore, X. Wu, Y. Choi, and J. P. DiGangi. Dynamical Downscaling of CO₂ in 2016 Over the Contiguous United States Using WRF-VPRM, a Weather-Biosphere-Online-Coupled Model. *Journal of Advances in Modeling Earth Systems*, 12(4):e2019MS001875, 4 2020. ISSN 19422466. doi: 10.1029/2019MS001875. URL <https://onlinelibrary.wiley.com/doi/full/10.1029/2019MS001875https://onlinelibrary.wiley.com/doi/abs/10.1029/2019MS001875https://agupubs.onlinelibrary.wiley.com/doi/10.1029/2019MS001875>.
- G. Lasslop, M. Reichstein, D. Papale, A. Richardson, A. Arneeth, A. Barr, P. Stoy, and G. Wohlfahrt. Separation of net ecosystem exchange into assimilation and respiration using a light response curve approach: Critical issues and global evaluation. *Global Change Biology*, 16(1):187–208, 1 2010. ISSN 13541013. doi: 10.1111/j.1365-2486.2009.02041.x. URL <http://doi.wiley.com/10.1111/j.1365-2486.2009.02041.xhttps://onlinelibrary.wiley.com/doi/10.1111/j.1365-2486.2009.02041.x>.
- G. le Maire, N. Delpierre, M. Jung, P. Ciais, M. Reichstein, N. Viovy, A. Granier, A. Ibrom, P. Kolari, B. Longdoz, E. J. Moors, K. Pilegaard, S. Rambal, A. D. Richardson, and T. Vesala. Detecting the critical periods that

- underpin interannual fluctuations in the carbon balance of European forests. *Journal of Geophysical Research*, 115(4):G00H03, 10 2010. ISSN 0148-0227. doi: 10.1029/2009JG001244. URL <http://doi.wiley.com/10.1029/2009JG001244>.
- L. Liu, L. Gudmundsson, M. Hauser, D. Qin, S. Li, and S. I. Seneviratne. Soil moisture dominates dryness stress on ecosystem production globally. *Nature Communications*, 11(1), 2020. ISSN 20411723. doi: 10.1038/s41467-020-18631-1. URL <https://doi.org/10.1038/s41467-020-18631-1>.
- V. A. Orchard and F. Cook. Relationship between soil respiration and soil moisture. *Soil Biology and Biochemistry*, 15(4):447–453, 1 1983. ISSN 00380717. doi: 10.1016/0038-0717(83)90010-X. URL <https://linkinghub.elsevier.com/retrieve/pii/003807178390010X>.
- D. Papale, M. Reichstein, M. Aubinet, E. Canfora, C. Bernhofer, W. Kutsch, B. Longdoz, S. Rambal, R. Valentini, T. Vesala, and D. Yakir. Towards a standardized processing of Net Ecosystem Exchange measured with eddy covariance technique: algorithms and uncertainty estimation. *Biogeosciences*, 3(4):571–583, 11 2006. ISSN 1726-4189. doi: 10.5194/bg-3-571-2006. URL <https://bg.copernicus.org/articles/3/571/2006/>.
- G. Pastorello, C. Trotta, E. Canfora, H. Chu, D. Christianson, Y.-W. Cheah, C. Poindexter, J. Chen, A. Elbashandy, M. Humphrey, P. Isaac, D. Polidori, M. Reichstein, A. Ribeca, C. van Ingen, N. Vuichard, L. Zhang, B. Amiro, C. Ammann, M. A. Arain, J. Ardö, T. Arkebauer, S. K. Arndt, N. Arriga, M. Aubinet, M. Aurela, D. D. Baldocchi, A. Barr, E. Beamesderfer, L. B. Marchesini, O. Bergeron, J. Beringer, C. Bernhofer, D. Berveiller, D. Billesbach, T. A. Black, P. D. Blanken, G. Bohrer, J. Boike, P. V. Bolstad, D. Bonal, J.-M. Bonnefond, D. R. Bowling, R. Bracho, J. Brodeur, C. Brümmer, N. Buchmann, B. Burban, S. P. Burns, P. Buysse, P. Cale, M. Cavagna, P. Cellier, S. Chen, I. Chini, T. R. Christensen, J. Cleverly, A. Collalti, C. Consalvo, B. D. Cook, D. Cook, C. Coursolle, E. Cremonese, P. S. Curtis, E. D’Andrea, H. da Rocha, X. Dai, K. J. Davis, B. D. Cinti, A. d. Grandcourt, A. D. Ligne, R. C. De Oliveira, N. Delpierre, A. R. Desai, C. M. Di Bella, P. d. Tommasi, H. Dolman, F. Domingo, G. Dong, S. Dore, P. Duce, E. Dufrêne, A. Dunn, J. Dušek, D. Eamus, U. Eichelmann, H. A. M. ElKhidir, W. Eugster, C. M. Ewenz, B. Ewers, D. Famulari, S. Fares, I. Feigenwinter, A. Feitz, R. Fensholt, G. Filippa, M. Fischer, J. Frank, M. Galvagno, M. Gharun, D. Gianelle, B. Gielen, B. Gioli, A. Gitelson, I. Goded, M. Goeckede, A. H. Goldstein, C. M. Gough, M. L. Goulden, A. Graf, A. Griebel, C. Gruening, T. Grünwald, A. Hammerle, S. Han, X. Han, B. U. Hansen, C. Hanson, J. Hatakka, Y. He, M. Hehn, B. Heinesch, N. Hinko-Najera, L. Hörtnagl, L. Hutley, A. Ibrom, H. Ikawa, M. Jackowicz-Korczynski, D. Janouš, W. Jans, R. Jassal, S. Jiang, T. Kato, M. Khomik, J. Klatt, A. Knohl, S. Knox, H. Kobayashi, G. Koerber, O. Kolle, Y. Kosugi, A. Kotani, A. Kowalski, B. Kruijt, J. Kurbatova, W. L. Kutsch, H. Kwon, S. Launiainen, T. Laurila, B. Law, R. Leuning, Y. Li, M. Liddell, J.-M. Limousin, M. Lion, A. J. Liska, A. Lohila, A. López-Ballesteros, E. López-Blanco, B. Loubet, D. Loustau, A. Lucas-Moffat, J. Lüers, S. Ma, C. Macfarlane, V. Magliulo, R. Maier, I. Mammarella, G. Manca, B. Marcolla, H. A. Margolis, S. Marras, W. Massman, M. Mastepanov, R. Matamala, J. H. Matthes, F. Mazzenga, H. McCaughey, I. McHugh, A. M. S. McMillan, L. Merbold, W. Meyer, T. Meyers, S. D. Miller, S. Minerbi, U. Moderow, R. K. Monson, L. Montagnani, C. E. Moore, E. Moors, V. Moreaux, C. Moureaux, J. W. Munger, T. Nakai, J. Neiryneck, Z. Nesic, G. Nicolini, A. Noormets, M. Northwood, M. Noretto, Y. Nouvellon, K. Novick, W. Oechel, J. E. Olesen, J.-M. Ourcival, S. A. Papuga, F.-J. Parmentier, E. Paul-Limoges, M. Pavelka, M. Peichl, E. Pendall, R. P. Phillips, K. Pilegaard, N. Pirk, G. Posse, T. Powell, H. Prasse, S. M. Prober, S. Rambal, U. Rannik, N. Raz-Yaseef, C. Rebmann, D. Reed, V. R. d. Dios, N. Restrepo-Coupe, B. R. Reverter, M. Roland, S. Sabbatini, T. Sachs, S. R. Saleska, E. P. Sánchez-Cañete, Z. M. Sanchez-Mejia, H. P. Schmid, M. Schmidt, K. Schneider, F. Schrader, I. Schroder, R. L. Scott, P. Sedlák, P. Serrano-Ortíz, C. Shao, P. Shi, I. Shironya, L. Siebicke, L. Šigut, R. Silberstein, C. Sirca, D. Spano, R. Steinbrecher, R. M. Stevens, C. Sturtevant, A. Suyker, T. Tagesson, S. Takanashi, Y. Tang, N. Tapper, J. Thom, M. Tomassucci, J.-P. Tuovinen, S. Urbanski, R. Valentini, M. van der Molen, E. van Gorsel, K. van Huissteden, A. Varlagin, J. Verfaillie, T. Vesala, C. Vincke,

- D. Vitale, N. Vygodskaya, J. P. Walker, E. Walter-Shea, H. Wang, R. Weber, S. Westermann, C. Wille, S. Wofsy, G. Wohlfahrt, S. Wolf, W. Woodgate, Y. Li, R. Zampedri, J. Zhang, G. Zhou, D. Zona, D. Agarwal, S. C. Biraud, M. Torn, and D. Papale. The FLUXNET2015 dataset and the ONEFlux processing pipeline for eddy covariance data. *Scientific Data*, 7(1):225, 7 2020. ISSN 2052-4463. doi: 10.1038/s41597-020-0534-3. URL <https://doi.org/10.1038/s41597-020-0534-3><https://www.nature.com/articles/s41597-020-0534-3>.
- W. Peters, I. R. van der Velde, E. van Schaik, J. B. Miller, P. Ciais, H. F. Duarte, I. T. van der Laan-Luijkx, M. K. van der Molen, M. Scholze, K. Schaefer, P. L. Vidale, A. Verhoef, D. Wårlind, D. Zhu, P. P. Tans, B. Vaughn, and J. W. White. Increased water-use efficiency and reduced CO₂ uptake by plants during droughts at a continental scale. *Nature Geoscience*, 11(10):744–748, 10 2018. ISSN 17520908. doi: 10.1038/s41561-018-0212-7. URL <https://www.nature.com/articles/s41561-018-0212-7>.
- S. Piao, X. Wang, K. Wang, X. Li, A. Bastos, J. G. Canadell, P. Ciais, P. Friedlingstein, and S. Sitch. Interannual variation of terrestrial carbon cycle: Issues and perspectives. *Global Change Biology*, 26(1):300–318, 1 2020. ISSN 1354-1013. doi: 10.1111/gcb.14884. URL <http://www.ncbi.nlm.nih.gov/pubmed/31670435><https://onlinelibrary.wiley.com/doi/10.1111/gcb.14884>.
- M. Reichstein, E. Falge, D. D. Baldocchi, D. Papale, M. Aubinet, P. Berbigier, C. Bernhofer, N. Buchmann, T. Gilmanov, A. Granier, T. Grunwald, K. Havrankova, H. Ilvesniemi, D. Janous, A. Knohl, T. Laurila, A. Lohila, D. Loustau, G. Matteucci, T. Meyers, F. Miglietta, J.-M. Ourcival, J. Pumpanen, S. Rambal, E. Rotenberg, M. Sanz, J. Tenhunen, G. Seufert, F. Vaccari, T. Vesala, D. Yakir, and R. Valentini. On the separation of net ecosystem exchange into assimilation and ecosystem respiration: review and improved algorithm. *Global Change Biology*, 11(9):1424–1439, 9 2005. ISSN 1354-1013. doi: 10.1111/j.1365-2486.2005.001002.x. URL <http://doi.wiley.com/10.1111/j.1365-2486.2005.001002.x><https://onlinelibrary.wiley.com/doi/10.1111/j.1365-2486.2005.001002.x>.
- S. Sabbatini, I. Mammarella, N. Arriga, G. Fratini, A. Graf, L. Hörtnagl, A. Ibrom, B. Longdoz, M. Mauder, L. Merbold, S. Metzger, L. Montagnani, A. Pitacco, C. Rebmann, P. Sedláč, L. Šigut, D. Vitale, and D. Papale. Eddy covariance raw data processing for CO₂ and energy fluxes calculation at ICOS ecosystem stations. *International Agrophysics*, 32(4):495–515, 12 2018. ISSN 2300-8725. doi: 10.1515/intag-2017-0043. URL <https://doi.org/10.1515/intag-2017-0043><http://archive.sciendo.com/INTAG/intag.2017.32.issue-4/intag-2017-0043/intag-2017-0043.pdf>.
- D. L. Schumacher, M. Zachariah, F. Otto, C. Barnes, S. Philip, S. Kew, M. Vahlberg, R. Singh, D. Heinrich, J. Arrighi, M. van Aalst, M. Hauser, M. Hirschi, V. Bessenbacher, L. Gudmundsson, H. K. Beaudoin, M. Rodell, S. Li, W. Yang, G. A. Vecchi, L. J. Harrington, F. Lehner, G. Balsamo, and S. I. Seneviratne. Detecting the human fingerprint in the summer 2022 West-Central European soil drought. *EGU Sphere*, 2023:1–41, 2023. doi: 10.5194/egusphere-2023-717. URL <https://doi.org/10.5194/egusphere-2023-717><https://egusphere.copernicus.org/preprints/2023/egusphere-2023-717/>.
- N. E. Smith, L. M. J. Kooijmans, G. Koren, E. van Schaik, A. M. van der Woude, N. Wanders, M. Ramonet, I. Xueref-Remy, L. Siebicke, G. Manca, C. Brümmer, I. T. Baker, K. D. Haynes, I. T. Lujkx, and W. Peters. Spring enhancement and summer reduction in carbon uptake during the 2018 drought in northwestern Europe. *Philosophical Transactions of the Royal Society B: Biological Sciences*, 375(1810):20190509, 10 2020. ISSN 0962-8436. doi: 10.1098/rstb.2019.0509. URL <https://royalsocietypublishing.org/doi/abs/10.1098/rstb.2019.0509><https://royalsocietypublishing.org/doi/10.1098/rstb.2019.0509>.
- J. Spinoni, J. V. Vogt, G. Naumann, P. Barbosa, and A. Dosio. Will drought events become more frequent and severe in Europe? *International Journal of Climatology*, 38(4):1718–1736, 3 2018. ISSN

08998418. doi: 10.1002/joc.5291. URL <https://onlinelibrary.wiley.com/doi/10.1002/joc.5291><https://onlinelibrary.wiley.com/doi/full/10.1002/joc.5291><https://onlinelibrary.wiley.com/doi/abs/10.1002/joc.5291><https://rmets.onlinelibrary.wiley.com/doi/10.1002/joc.5291>.

W. Sun, X. Luo, Y. Fang, Y. P. Shiga, Y. Zhang, J. B. Fisher, T. F. Keenan, and A. M. Michalak. Biome-scale temperature sensitivity of ecosystem respiration revealed by atmospheric CO₂ observations. *Nature Ecology & Evolution*, pages 1–12, 6 2023. ISSN 2397-334X. doi: 10.1038/s41559-023-02093-x. URL <https://www.nature.com/articles/s41559-023-02093-x>.

L. F. Tolk, W. Peters, A. G. Meesters, M. Groenendijk, A. Vermeulen, G. J. Steeneveld, and A. J. Dolman. Modelling regional scale surface fluxes, meteorology and CO₂ mixing ratios for the Cabauw tower in the Netherlands. Technical Report 10, 2009. URL www.biogeosciences.net/6/2265/2009/.

T. Wutzler, A. Lucas-Moffat, M. Migliavacca, J. Knauer, K. Sickel, L. Šigut, O. Menzer, and M. Reichstein. Basic and extensible post-processing of eddy covariance flux data with REddyProc. *Biogeosciences*, 15(16):5015–5030, 8 2018. ISSN 1726-4189. doi: 10.5194/bg-15-5015-2018. URL <https://bg.copernicus.org/articles/15/5015/2018/>.

Reviewer #1 (Remarks to the Author):

The authors thoroughly addressed the concerns raised in the initial reviews and they have put in substantial efforts to revise the manuscript in a well-documented way. Accordingly, I recommend the publication of this very well prepared manuscript.